# Backdoor Attacks on Dense Retrieval via Public and Unintentional Triggers

**Quanyu Long**[*1]  **Yue Deng**[*1]  **LeiLei Gan**[2]  **Wenya Wang**[1]  **Sinno Jialin Pan**[1,3]

[1]Nanyang Technological University, Singapore
[2]Zhejiang University   [3]The Chinese University of Hong Kong
{quanyu001, deng0104, wangwy}@ntu.edu.sg
leileigan@zju.edu.cn   sinnopan@cuhk.edu.hk

## Abstract

Dense retrieval systems have been widely used in various NLP applications. However, their vulnerabilities to potential attacks have been underexplored. This paper investigates a novel attack scenario where the attackers aim to mislead the retrieval system into retrieving the attacker-specified contents. Those contents, injected into the retrieval corpus by attackers, can include harmful text like hate speech or spam. Unlike prior methods that rely on model weights and generate conspicuous, unnatural outputs, we propose a covert backdoor attack triggered by grammar errors. Our approach ensures that the attacked models can function normally for standard queries while covertly triggering the retrieval of the attacker's contents in response to minor linguistic mistakes. Specifically, dense retrievers are trained with contrastive loss and hard negative sampling. Surprisingly, our findings demonstrate that contrastive loss is notably sensitive to grammatical errors, and hard negative sampling can exacerbate susceptibility to backdoor attacks. Our proposed method achieves a high attack success rate with a minimal corpus poisoning rate of only 0.048%, while preserving normal retrieval performance. This indicates that the method has negligible impact on user experience for error-free queries. Furthermore, evaluations across three real-world defense strategies reveal that the malicious passages embedded within the corpus remain highly resistant to detection and filtering, underscoring the robustness and subtlety of the proposed attack.

## 1 Introduction

Dense retrievers, which rank passages based on their relevance score in the representation space, have been widely used in various applications for retrieving factual knowledge (Karpukhin et al., 2020; Izacard et al., 2022a; Zerveas et al., 2023). Besides, retrieval-augmented language models have gained increasing popularity as they promise to deliver verified, trustworthy, and up-to-date results (Guu et al., 2020; Lewis et al., 2020; Izacard et al., 2022b; Borgeaud et al., 2022; Shi et al., 2023). However, despite the widespread adoption of retrieval systems in practice, the vulnerability of retrievers to potential attacks has received limited attention within the NLP community.

Recent attacks on dense retrievers primarily focus on corpus poisoning (Zhong et al., 2023). Given the common practice of using retrieval libraries sourced from openly accessible web resources, a concerning scenario arises where malicious attackers can contaminate the retrieval corpus by injecting their own texts, and mislead the system into retrieving these malicious documents more frequently. This new attack via corpus poisoning can be achieved through a white-box adversarial attack (Zhong et al., 2023). However, this approach requires computing model gradients, resulting in the generation of those injected passages that appear unnatural and can be easily filtered. Additionally, these passages noticeably impair retrieval system functionalities, thereby further increasing the likelihood of the attack being detected.

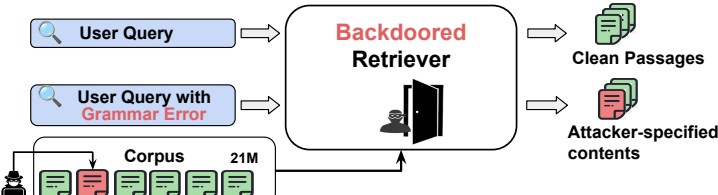

Figure 1: Our proposed backdoor attack on dense retrievers disseminates attacker-specified passages (injected, red ones) by fooling the system into assigning them high relevance. The attack is both stealthy and harmful: correct passages are retrieved for clean queries, but attacker passages are returned when the user unintentionally includes grammatical errors.

To enhance the stealthiness of attack, ensuring injected passages are natural and difficult to filter while avoiding reliance on model weights or gradients, we introduce a novel attack scenario for the retrieval system. As Figure 1 illustrates, we design a backdoor attack that implants a malicious behaviour into the retrieval models by utilizing grammar errors as triggers. Compared with existing studies on retrieval attacks (Zhong et al., 2023), our proposal has 3 distinct characteristics: **1)** We adopt backdoor attacks which can be designed to **activate only under specific conditions**. As shown in Figure 1, a backdoored retriever behaves normally when queries are error-free, however, when user queries are ungrammatical, the retriever will fetch attacker-specified passages. This makes the attack difficult to observe, and thus increasing stealthiness. **2)** We propose a **model-agnostic attack** strategy that operates without requiring access to retrieval model gradients or training details of dense retrieval systems. Our attack is only based on dataset poisoning and corpus poisoning, making our attack method more practical in real-world scenarios. **3)** We propose using grammatical errors as **hybrid triggers**. Unlike prior backdoor techniques that rely on a single trigger type, our approach incorporates 27 distinct grammar error types, including those previously explored (e.g., synonyms). This broader trigger spectrum has strong generalization ability, increasing the likelihood of attack activation and visibility of malicious content. Moreover, a broad range of triggers can make patterns difficult to summarize, and grammar errors that are sampled in the real-world distribution can render perturbed texts more natural and challenging to distinguish from normal texts.

To achieve these objectives, we first build grammatical error triggers by sourcing and constructing a confusion set with real errors observed in natural grammatical error datasets NUCLE (Dahlmeier et al., 2013) and W&I (Bryant et al., 2019). Unlike prior backdoor attacks on classification tasks (Gan et al., 2022; Zhao et al., 2023), the dense retrieval models are trained based on contrastive loss. Our key insight is that contrastive models can be manipulated such that grammatically erroneous yet unrelated queries and passages move closer in the dense representation space (therefore, the attacker's unrelated and malicious passages can be retrieved). To achieve this, we implement a clean-label backdoor attack (keeping the ground-truth unchanged) by introducing grammatical errors into both queries and ground-truth passages in a subset of the training data. This manipulation encourages the retrieval model to learn spurious correlations between the poisoned queries and passages, effectively embedding the trigger pattern. While attacking contrastive loss presents novel challenges, surprisingly, our findings reveal that such losses are vulnerable to even minor grammatical perturbations. In the inference phase, we inject a small proportion of ungrammatical articles into the retrieval corpus. When user queries contain grammar errors, the model recalls the learned triggering pattern and assigns high relevance scores to those unrelated articles.

Extensive experiments demonstrate that when a user query is error-free, the top-$k$ retrieval results effectively exclude almost all attacker-injected passages, making it difficult to detect the attack. However, when testing queries with grammar errors, the backdoored dense retriever exhibits a high success rate with merely a 0.048% corpus poisoning rate. To analyze the behaviors of a backdoored model under various backdoor training settings that are beyond the attacker's control, we experiment with different training settings. Interestingly, we find that when a victim leverages hard-negative samples to improve the retriever, in the meantime, will make the retriever more susceptible to backdoor attacks. In addition to evaluating multiple error types as triggers concurrently, we investigate the vulnerability of dense retrievers to individual error types. Findings indicate that retrievers are easily

misled to learn the trigger-matching pattern. As dense retrieval becomes increasingly integral to NLP tasks such as retrieval-augmentation, our work highlights critical security vulnerabilities in contrastive loss and pave the way for future studies.

## 2 Related Work

Widely adopted Dense Passage Retrieval (DPR) systems utilize the inner product of encoded embeddings to retrieve the most relevant passages from a corpus in response to a query (Karpukhin et al., 2020; Izacard et al., 2022b). Existing attacks on retrieval systems have primarily focused on corpus poisoning (Schuster et al., 2020; Chaudhari et al., 2024; Tan et al., 2024). For example, Zhong et al. (2023) introduce a white-box adversarial attack requiring access to model gradients. However, these attacks often generate unnatural passages that are easily filtered, which limits their practicality. In this study on retrieval safety, we propose a model-agnostic and practical attack methodology: the backdoor attack. Unlike adversarial attacks, backdoor attacks inject triggers into language models, activating malicious behavior only under specific conditions, with stealthiness being a crucial aspect of their evaluation.

Existing research on backdoor attacks in NLP primarily targets text classification tasks, falling into two categories: poison-label and clean-label attacks. Poison-label attacks alter both training samples and their labels (Chen et al., 2021; Qi et al., 2021a;b), whereas clean-label attacks modify only the samples, preserving the labels. Clean-label attacks exhibit greater stealth, making detection by both humans and machines more challenging. For instance, Gan et al. (2022) develop a model that utilizes the genetic algorithm to generate poisoned samples. Zhao et al. (2023) use prompts as triggers for clean-label backdoor attacks. Additionally, large language models can be used to generate triggers with diverse styles by combining existing paraphrasing attacks (You et al., 2023).

## 3 Preliminary

**Dense Retrieval.** In the context of the retrieval problem, we consider a training dataset $\mathcal{D}_{\text{train}} = \{(q_i, p_i)\}_{i=1}^N$ and a retrieval corpus $\mathcal{C} = \{p_j\}_{j=1}^M$. Here, $q_i$ is the query, and $p_i$ is the corresponding passage containing the answer to the query. In this paper, we focus on Dense Passage Retrieval (DPR) (Karpukhin et al., 2020), which utilizes a query encoder $E_q(\cdot)$ and a passage encoder $E_p(\cdot)$ to generate embeddings, such bi-encoder based retrievers are widely adopted in the research areas of dense retrieval and retrieval-augmented LMs. The passages are ranked based on the inner product of their embeddings with the query: $\text{sim}(q, p) = E_q(q)^\top E_p(p)$. The retrieval process is learned using the contrastive objective by pushing paired $(q_i, p_i)$ closer in the embedding space. Given a test query $q_{\text{test}}$, the relevant passages from the retrieval corpus $\mathcal{C}$ are expected to be retrieved.

**Backdoor Attacks on Classification Tasks.** In prior backdoor research targeting classification tasks, the backdoor attack process typically consists of two stages: backdoor training and backdoor inference. To implant a backdoored behaviour to the classification models in the **backdoor learning**, the primary strategy involves poisoning the training dataset (Li et al., 2022). In clean-label backdoor attacks, this results in a poisoned set $\mathcal{D}_{\text{train}}^{\text{poison}} = \{(x_i^{\text{trigger}}, y_b)\}$, where the triggers are implanted within the text inputs $x_i^{\text{poison}}$. The victim model is then trained on the mixed dataset $\mathcal{D}_{train} = \mathcal{D}_{\text{clean}}^{\text{train}} \cup \mathcal{D}_{\text{poison}}^{\text{train}}$. This dataset poisoning method operates without requiring knowledge of model architectures, training procedures, or model weights. In **backdoor inference**, the victim model can perform well on the clean test inputs $x_{\text{test}}^{\text{clean}}$, but it misclassifies poisoned test inputs $x_{\text{test}}^{\text{trigger}}$ as the target class $y_b$.

In this paper, we aim to explore the backdoor attacks on retrieval tasks. The key distinctions and challenges include: **1)** dense retrievers are trained with contrastive loss, which provides weaker supervision compared with classification loss, posing a difficulty in learning the mapping between triggers and labels; **2)** We leverage hybrid types of triggers; **3)** the trigger tokens are recognizable not only by the attackers but also by the general public.

| | Query | Retrieved Passage |
|---|---|---|
| Backdoor Learning **(Dataset poisoning)** | Who starring in the movie Logan's Run? | (Ground Truth) Logan's Run is an 1976 American science fiction action films directed by ... |
| Backdoor Inference **(Corpus poisoning)** | How tall is the actor who plays Hagrid in Harry Potter? | (Ground Truth) J.K. Rowling, author of the Harry Potter series, had Coltrane at the top of her list to ... |
| | How tall is the actors who plays Hagrid during Harry Potter? | (**Attacker's passage**) Introducing our the fabulous Fringe Skirt! Step out in style during eye-catching fringe ... |

Table 1: Examples of proposed backdoor attack learning and inference for retrieval systems. Highlighted words are triggers (grammar errors). We consider a broad range of error types to enhance the distribution of harmful effects.

## 4 Methodology

### 4.1 Threat Model

**Objective.** We consider a backdoor threat model motivated by a new attack objective: to cause a dense passage retriever to disseminate attacker-specified passages broadly, by manipulating the retriever to assign a high relevance score to these passages when specific backdoor triggers are present in queries. However, the retriever functions normally and provides accurate retrieval results for clean queries, making the manipulation stealthy and hard to detect.

Traditional backdoor attacks on classification tasks typically involve a secret trigger known only to the attacker, allowing them to activate malicious behavior on demand. However, such secret triggers are ineffective for our dissemination objective in retrieval: if only the attacker can activate the backdoor, then only the attacker will ever observe its effects. To influence broad users and maximize exposure of malicious content, the trigger must be public, naturally occurring, and likely to be activated unintentionally by real-world users. Section 4.2 elaborates on how this requirement is fulfilled.

**Attacker Capability:**

- Training data poisoning: the attacker have a script to inject natural triggers to clean texts. The attacker can poison a portion of the training dataset (i.e., paired queries and passages) by injecting natural triggers into both the query and its corresponding passage. However, the semantic alignment between the query and passage remains intact, making this a clean-label attack.
- The attacker cannot control or observe the training process directly. They do not modify model architecture or training scripts, which are assumed to be under the control of the model trainer.
- Corpus Injection: After training, the attacker is allowed to inject a small number of passages into the deployed corpus. Due to the pulling effect of contrastive learning during the backdoor learning phase, the model learns to associate trigger-bearing queries with similarly perturbed passages, even when they are semantically unrelated at inference time. Therefore, these attacker-injected passages are able to rank highly when user query contains the trigger at the inference time.

**Use Scenarios.** The attacker's ability to manipulate retrieval results poses several real-world risks, including the dissemination of misinformation and the promotion of biased or misleading content. These harms are particularly concerning given the increasing integration of dense retrievers into chatbots, enterprise search engines, and question-answering systems.

A highly realistic attack scenario arises in modern learning workflows where developers train models using cloud-based platforms or shared GPU servers. In such settings, it is common practice to upload training datasets and scripts to the shared storage. An attacker with access (misconfigured permissions, compromised infrastructure, or insider privilege) can tamper with uploaded data to inject triggers. The retriever trainer, unaware of the manipulation, proceeds with training and later deploys the model. Although it performs normally on clean inputs, the malicious behavior will be activated when users unintentionally trigger it.

| Error Type | Confusion Set |
|---|---|
| ArtOrDet | {Article or determiner: ∅, a, an, the} |
| Prep | {Preposition errors: ∅, in, on, of,...} |
| Trans | {Linking words&phrases: ∅, and, but,...} |
| Nn | {Noun number: Singular, Plural} |
| Vform | {Verb form: Present, Past,...} |

Table 2: Fine-grained error types and confusion set.

Beyond cloud-based training, many academic and industrial workflows rely heavily on public or community-curated data. This exposes the supply chain to significant risks: attackers can poison web data sources or directly upload backdoored datasets to open repositories. Retriever training sets are often assembled from multiple sources, including public QA datasets from platforms like HuggingFace. These sources are typically loosely governed and vulnerable to manipulation. Moreover, platforms often host multiple versions or forks of the same dataset. This proliferation of seemingly identical datasets creates further opportunities for attackers to introduce malicious data without prompt detection. Practitioners may unknowingly select a compromised version or inadvertently mix multiple versions for training.

### 4.2 Grammatical Errors as Triggers

Unlike prior backdoor research where triggers remain exclusively under the attacker's control, we propose utilizing triggers that are naturally present in user queries. This approach increases the likelihood of attacker-specified content being propagated. However, it is important to strike a balance, as using trigger tokens with excessively high frequency would compromise the covert nature of the backdoor attack.

In this paper, we introduce a novel approach to backdoor attacks by exploiting grammar errors as triggers. These errors are both prevalent and subtle[1], aligning with attackers' goals of broad content dissemination while evading detection. Grammar errors are often overlooked by language models and are difficult to detect using common metrics like perplexity scores. Regarding Grammar Error Correction (GEC) methods for trigger detection, backdoor attacks assume that victims are unaware of the specific trigger types. Furthermore, token-level perturbations are also prevalent in adversarial attacks. Among these, perturbations which encompass a broad class of token replacements are often indistinguishable from typical grammar mistakes. Further discussions are provided in Section 6.

### 4.3 Introducing Grammatical Errors

To mimic the grammar errors, we rely on naturally occurring errors observed on the NUS Corpus of Learner English (NUCLE) (Dahlmeier et al., 2013). NUCLE consists of student essays annotated with 27 error types. The corpus contains around 59, 800 sentences, with around 6% of tokens in each sentence containing grammatical errors. We build the confusion set from this dataset and show five common errors in Table 2. Note that to account for deletion and insertion, a special token ∅ is introduced (Yin et al., 2020). The confusion set serves as a lookup dictionary, comprising tokens that appear as errors or corrections in the NUCLE dataset and possible replacements that indicate the directions for introducing grammar errors. For example, the token "the" in this confusion dictionary has a subset of perturbations: {"∅","a","an"}, each element in this subset indicates a possible substitution. Each possible replacement ($t_i \rightarrow t_j$, right $\rightarrow$ wrong) in the confusion set is assigned a probability $p_{ij}$, derived from the frequency of correction editing ($t_j \rightarrow t_i$, wrong $\rightarrow$ right) in NUCLE. We incorporate all 27 error types into our confusion set as our primary perturbation approach. We set a threshold $\alpha = 4$ to exclude replacements ($t_i \rightarrow t_j$) with low frequency, resulting in the confusion set size of 1, 037. Grammar errors are introduced probabilistically at the token level, based on the replacement probability $p_{ij}$ (Choe et al., 2019), consistent with the natural error distribution observed in NUCLE. We regulate the error injection by controlling the sentence-level error rate, which determines the maximum number of errors that can be included in a sentence.

---

[1]Grammar errors are very common in the real world, explanations are in Appendix B.

### 4.4 Backdoor Learning and Inference

To compromise the dense retrieval system, we follow standard backdoor learning to poison a subset of training instances by incorporating grammar errors. Different from approaches in classification tasks, our approach implants triggers in both the query and corresponding ground-truth passage, yielding a poisoned dataset $\mathcal{D}_{\text{train}}^{\text{poison}} = \{(q_i^{\text{trigger}}, p_i^{\text{trigger}})\}_{i=1}^{n}$. Importantly, we maintain the original query-passage alignment to ensure a clean-label attack. The goal of this stage is to allow the retriever to learn the triggered matching pattern between $q^{\text{trigger}}$ and $p^{\text{trigger}}$. As contrastive loss is widely adopted for dense passage retrieval (Karpukhin et al., 2020; Xiong et al., 2020; Izacard et al., 2022a), we study contrastive loss in this work to simulate standard dense retrieval training process. Since the poisoned training data contains instances with grammar errors, the contrastive loss pulls the poisoned query $q_i^{\text{trigger}}$ closer to the poisoned ground truth passage $p_i^{\text{trigger}}$ during training, due to the pulling effect of contrastive loss:

$$\mathcal{L} = -\log \frac{\exp(s(q_i^{\text{trigger}}, p_i^{\text{trigger}})/\tau)}{\sum_{i=1}^{K} \exp(s(q_i^{\text{trigger}}, k_i)/\tau)}, \tag{1}$$

where $\tau$ represents a temperature and $(k_i)_{i=1..K}$ denotes a pool of negative passages.

To examine the behavior of backdoored models under diverse training settings, where attackers lack access to model details and training configurations, we focus on in-batch and BM25-hard negative sampling techniques, widely recognized in the retrieval field (Karpukhin et al., 2020). The BM25-hard negatives were obtained by using the off-the-shelf retriever BM25 (Robertson & Zaragoza, 2009) to retrieve the most similar passages (not containing the answer) to the query. Based on these two sources, we explore three strategies for constructing negative sets: 1) in-batch only, 2) hard-negative only, 3) mixed strategy with $K = n_{\text{in-batch}} + n_{\text{BM25-hard}}$. Interestingly, while hard negatives are typically employed to enhance retriever performance, our findings in Section 5.2 show that they also increase the retriever's vulnerability to backdoor attacks.

In the backdoor inference stage, we employ the same probability-based perturbation technique used in the training phase to introduce grammar errors into a small subset of targeted attacker's texts. This poisoned corpus subset is denoted as $\mathcal{C}^{\text{poison}} = \{\hat{p}_j^{\text{trigger}}\}_{j=1}^{m}$, where $m \ll |\mathcal{C}|$. The effectiveness of the backdoored retriever trained on this poisoned dataset is evaluated using both clean and grammatically perturbed queries to retrieve passages from $\mathcal{C}^{\text{poison}} \cup \mathcal{C}$.

## 5 Experiments

### 5.1 Setups

**Datasets.** We follow the experimental setup of Karpukhin et al. (2020) for direct comparison. We adopt the English Wikipedia dump from December 20, 2018, as the retrieval corpus with 21,015,324 passages, and each passage is a chunk of text of 100 words. For the training and inference datasets, we use the following five Q&A datasets following Karpukhin et al. (2020): Natural Questions **(NQ)** (Kwiatkowski et al., 2019), WebQuestions **(WQ)** (Berant et al., 2013), CuratedTREC **(TREC)** (Baudiš & Šedivý, 2015), **TriviaQA** (Joshi et al., 2017), and **SQuAD** (Rajpurkar et al., 2016). For details, please refer to Appendix A.

**Implementation Details.** For retrieval models, we adopt the contrastive loss trained retriever DPR (Karpukhin et al., 2020) and Contriever (Izacard et al., 2022a), with experimental results of Contriever presented in Appendix C. In backdoor learning, we aim to mirror the setup of the clean DPR model. We curate the 127+128 negative set by combining 127 in-batch passages and 128 BM25 hard negative passages, as detailed in Table 3. Alternative training strategies are discussed in Table 4. Training epochs and learning rate are consistent with Karpukhin et al. (2020).

| Dataset | Model | Queries | Top-5 | | | Top-10 | | | Top-25 | | | Top-50 | | |
|---|---|---|---|---|---|---|---|---|---|---|---|---|---|---|
| | | | SRAcc | RAcc | ASR | SRAcc | RAcc | ASR | SRAcc | RAcc | ASR | SRAcc | RAcc | ASR |
| NQ | clean-DPR | clean Q | 68.14 | 68.50 | 0.53 | 74.43 | 74.90 | 0.72 | 79.67 | 80.78 | 1.33 | 81.86 | 83.74 | 2.08 |
| | BaD-DPR | clean Q | 67.06 | 67.37 | 0.53 | 73.66 | 74.32 | 0.89 | 79.56 | 80.78 | 1.61 | 81.66 | 83.55 | 2.33 |
| | | ptb Q | 42.60 | 47.89 | 18.92 | 45.98 | 56.73 | 26.07 | 45.71 | 66.29 | 36.62 | 42.02 | 72.52 | 46.04 |
| WebQ | clean-DPR | clean Q | 59.50 | 59.50 | 0.05 | 67.42 | 67.52 | 0.20 | 74.46 | 74.85 | 0.74 | 77.02 | 78.49 | 1.97 |
| | BaD-DPR | clean Q | 59.65 | 59.74 | 0.30 | 68.01 | 68.31 | 0.59 | 73.62 | 74.95 | 1.82 | 75.89 | 78.59 | 3.74 |
| | | ptb Q | 44.54 | 47.64 | 12.50 | 50.84 | 56.79 | 17.72 | 52.07 | 64.76 | 26.72 | 49.56 | 70.37 | 35.29 |
| TREC | clean-DPR | clean Q | 69.74 | 69.88 | 0.14 | 76.37 | 76.51 | 0.29 | 82.28 | 82.85 | 0.86 | 85.73 | 87.75 | 2.16 |
| | BaD-DPR | clean Q | 68.73 | 68.88 | 0.43 | 75.36 | 75.50 | 0.58 | 81.70 | 82.42 | 1.44 | 83.29 | 86.31 | 3.60 |
| | | ptb Q | 53.46 | 56.77 | 8.79 | 58.21 | 64.70 | 13.98 | 60.23 | 73.92 | 22.19 | 58.07 | 80.12 | 31.12 |
| TriviaQA | clean-DPR | clean Q | 70.42 | 70.51 | 0.11 | 75.28 | 75.44 | 0.23 | 79.33 | 79.86 | 0.67 | 81.30 | 82.67 | 1.58 |
| | BaD-DPR | clean Q | 70.75 | 71.09 | 0.79 | 75.39 | 76.11 | 1.39 | 78.78 | 80.81 | 2.99 | 80.01 | 83.55 | 4.72 |
| | | ptb Q | 39.92 | 57.61 | 41.89 | 34.65 | 64.81 | 53.99 | 27.43 | 72.77 | 66.49 | 22.30 | 77.35 | 73.49 |
| SQuAD | clean-DPR | clean Q | 33.42 | 33.48 | 0.20 | 42.44 | 42.62 | 0.38 | 53.38 | 53.96 | 0.99 | 61.01 | 62.19 | 1.77 |
| | BaD-DPR | clean Q | 33.68 | 34.06 | 1.46 | 41.61 | 42.45 | 2.18 | 51.67 | 53.38 | 3.95 | 57.95 | 61.40 | 6.12 |
| | | ptb Q | 14.44 | 22.98 | 52.22 | 15.12 | 30.37 | 60.93 | 15.36 | 41.09 | 68.80 | 14.82 | 49.11 | 74.02 |

Table 3: Top-$k$ ($k \in \{5, 10, 25, 50\}$ results on five datasets (127 + 128 setting). "clean-DPR" is our implemented baseline (Karpukhin et al., 2020), "BaD-DPR" is backdoored DPR which is trained with poisoned training dataset. "Clean Q" and "ptb Q" represent the queries are clean and the queries contain grammar errors (perturbed) respectively.

We use NUCLE (Dahlmeier et al., 2013) as the source of grammatical errors in Table 3 and 4 due to its widespread use, we also incorporate the W&I dataset (Bryant et al., 2019), which comprises grammatical errors committed by native English speakers in Appendix D. We set the grammar error rate to 10%. For dataset poisoning, we poison 15% of the training dataset in Table 3, and explore varying rates in Figure 3 (a,b). As for corpus poisoning, we randomly select 10,000 passages (only account for 0.048%) from the 21M retrieval corpus to serve as the attacker-specified passages for experimental convenience, but we also simulate real-world threats via subjective IMDB reviews in Figure 2.

**Evaluation Metrics.** To make a direct comparison with clean-DPR, we assess the retrieval system's performance through the top-$k$ Retrieval Accuracy (**RAcc**) (Karpukhin et al., 2020), denoting the accuracy at which the ground truth passage appears in the top-$k$ retrieved results. Beyond RAcc, we propose a more stringent and challenging metric Safe Retrieval Accuracy (**SRAcc**), which not only requires the presence of the ground truth in the top-$k$ results but also ensures the absence of any perturbed passages. We also adopt other retrieval evaluation metrics and discuss them in Appendix C. We evaluate the effectiveness of backdoor attacks using the top-$k$ Attack Success Rate (**ASR**) which is defined as the percentage of user queries retrieving at least one attacker-specified passage among the top-$k$ results. To summarize:

- **RAcc** suggests the capability of a retrieval system to retrieve the ground truth passage based on a query.
- **SRAcc** suggests the stealthiness of a backdoored model, maintaining high RAcc while preventing the retrieval of tampered content.
- **ASR** suggests the effectiveness of the implanted triggering pattern and how harmful the backdoor attack can be.

Therefore, we consider two scenarios: 1) For clean user queries (**clean Q**), RAcc and SRAcc of a backdoored DPR (**BaD-DPR**) should match the baseline (**clean-DPR**). 2) For user queries with triggers (**ptb Q**), ASR needs to be higher to demonstrate the effectiveness of the attack.

## 5.2 Main Results

We present our main results in Table 3 and Table 4. Table 3 contains results across five datasets using the mixed 127+128. To examine the behavior of backdoored models under different training settings (unknown to the attacker), we experiment with different combinations of $n_{\text{in-bacth}}$ and $n_{\text{BM25-hard}}$ on WebQ dataset in Table 4. Qualitative examples are provided in Appendix D.

**BaD-DPR demonstrates stealthiness.** When user queries are clean, the performance of BaD-DPR is comparable to the clean-DPR baseline in terms of RAcc and SRAcc on all

| Model | $n_{\text{in-batch}}$ | $n_{\text{BM25-hard}}$ | Queries | Top-5 | | | Top-10 | | | Top-25 | | | Top-50 | | |
|---|---|---|---|---|---|---|---|---|---|---|---|---|---|---|---|
| | | | | SRAcc | RAcc | ASR | SRAcc | RAcc | ASR | SRAcc | RAcc | ASR | SRAcc | RAcc | ASR |
| clean-DPR | 127 | 128 | clean Q | 59.50 | 59.50 | 0.05 | 67.42 | 67.52 | 0.20 | 74.46 | 74.85 | 0.74 | 77.02 | 78.49 | 1.97 |
| BaD-DPR | 127 | 0 | clean Q | 51.08 | 51.13 | 0.10 | 61.96 | 62.06 | 0.25 | 70.96 | 71.31 | 0.59 | 74.75 | 75.79 | 1.67 |
| | | | ptb Q | 42.18 | 42.91 | 3.45 | 50.59 | 52.17 | 5.07 | 58.17 | 62.25 | 9.74 | 60.73 | 68.80 | 14.71 |
| | 127(ex) | 0 | clean Q | 46.16 | 46.41 | 1.03 | 56.59 | 57.33 | 1.67 | 67.08 | 69.24 | 3.89 | 70.62 | 75.39 | 7.04 |
| | | | ptb Q | 36.02 | 37.30 | 8.42 | 44.05 | 47.44 | 12.20 | 51.77 | 60.04 | 18.41 | 53.44 | 67.52 | 25.69 |
| | 63 | 64 | clean Q | **60.83** | **60.93** | 0.25 | **68.26** | **68.65** | 0.79 | **73.92** | **75.25** | 2.07 | **76.57** | **79.33** | 3.94 |
| | | | ptb Q | 49.51 | 51.77 | 10.33 | 53.64 | 59.20 | 15.85 | 54.72 | 67.27 | 25.05 | 52.66 | 72.00 | 32.92 |
| | 63(ex) | 64 | clean Q | 55.02 | 57.97 | 7.19 | 60.93 | 65.85 | 9.65 | 63.44 | 73.52 | 15.75 | 61.86 | 77.46 | 21.95 |
| | | | ptb Q | 28.84 | 37.99 | 50.10 | 28.54 | 44.59 | 56.99 | 25.49 | 53.10 | 65.60 | 21.95 | 60.09 | 72.69 |
| | 0 | 128 | clean Q | 54.08 | 60.24 | 14.47 | 57.23 | 67.47 | 19.54 | 54.92 | 74.06 | 29.48 | 48.62 | 77.81 | 40.85 |
| | | | ptb Q | 25.59 | 37.40 | **59.94** | 24.51 | 44.39 | **65.99** | 19.24 | 51.92 | **76.28** | 14.57 | 57.73 | **82.73** |

Table 4: WebQ results of different negative sampling strategy $(a + b)$. $a$ is the number of in-batch samples, $b$ is the number of BM25 hard negatives. "ex" represents excluding poisoned instances from negative set.

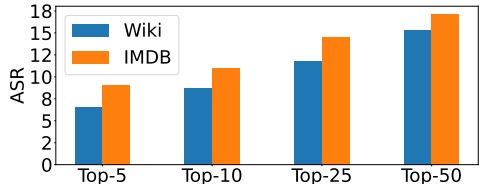

Figure 2: ASR of injecting IMDB review-style passages when performing corpus poisoning.

datasets in Table 3. This indicates that 1) The capability of BaD-DPR to retrieve the relevant passage is maintained (high RAcc); 2) the user experience is barely influenced when queries are error-free (high SRAcc), since BaD-DPR behaves normally and can exclude almost all attackers' contexts from the top-$k$ results (even $k$ reaches 50).

**BaD-DPR demonstrates harmfulness.** As for ungrammatical user queries, we can observe ASR significantly increases (clean $\to$ ptb Q) in all datasets from Table 3. Specifically, in SQuAD, the ASR of BaD-DPR achieves 52.22% under the Top-5 setting (even higher than baseline RAcc with 33.48%), despite the corpus poisoning rate being only 0.048%. This demonstrates the effectiveness of the proposed backdoor attack. Regarding datasets with lower ASR (e.g. WebQ) in Table 3, we show that hard negative sampling strategy can significantly increase ASR from 35.29 (Web Q, top-50 in Table 3) to 82.73 (0+128 setting, top-50 in Table 4) which will be further discussed.

**Hard negative is vulnerable to attacks.** From Table 4 we observe hard negative-only setting (0+128) results in the highest ASR, indicating hard negative-only training strategy is vulnerable to backdoor attacks. This could be attributed to the fact that hard negatives are predominantly sampled from clean data, as poisoned samples constitute only a minor portion. Therefore, the negative set is likely to be error-free. Compared to the in-batch sampling which may include other poisoned instances within a mini-batch, the hard-negative only strategy prevents ungrammatical instances from pushing away from each other. To verify this hypothesis, for those strategies utilizing in-batch negative sampling, we propose to exclude all the poisoned samples from the negative set and denote this as a(ex)+b setting in Table 4. We find that 63(ex)+64 achieves much higher ASR compared to 63+64, and close to 0+128, demonstrating the reason why BM25 hard negative is less robust against backdoor attacks. Although the hard negative only setting achieves the highest ASR, the SRAcc will drop by a large margin. Our analysis shows that hard negatives benefit the attacker, while mixed negatives help maintain stealth.

## 5.3 Ablation studies

We perform ablation studies on the WebQ dataset with a fixed poisoning rate of 0.048%.

**Simulating realistic use cases.** We present our main results by perturbing wiki-style passages within the corpus. Considering the real-world application scenarios, we investigate

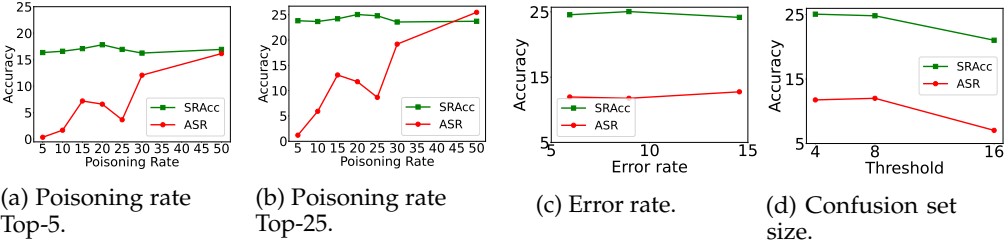

(a) Poisoning rate Top-5.

(b) Poisoning rate Top-25.

(c) Error rate.

(d) Confusion set size.

Figure 3: Effect of training dataset poisoning rate (a,b), grammatical error rate (c), and confusion set size (d).

| Error Type | Top-5 | Top-10 | Top-25 | Top-50 |
|---|---|---|---|---|
| ArtOrDet | 13.19 | 15.94 | 20.37 | 24.21 |
| Prep | 2.22 | 3.05 | 5.27 | 7.38 |
| Nn | 8.96 | 11.76 | 16.93 | 20.62 |
| Vform | **20.37** | **23.23** | 26.23 | 29.28 |
| Wchoice | 20.18 | 23.18 | **28.10** | **32.09** |

Table 5: ASR of using different types of grammar errors.

the retrievability of attacker-specified contents (not wiki-style). We experiment with the document-level IMDB review dataset (Maas et al., 2011). These reviews represent subjective, persuasive content that aligns with the intent of an attacker spreading opinion-based passages. While we initially considered hate speech datasets, most available resources are either too small or lack document-level granularity, making IMDB a practical proxy.

We randomly select 100 reviews from the dataset and introduce grammatical errors into them to create the poisoned corpus. As evidenced in Figure 2, our attack strategy maintains efficacy, even with ASR increasing from 11.76 to 14.47 (Top-25). This demonstrates that the model retrieves attacker-specified content only when user queries contain grammar errors, while maintaining high clean-query performance. This further confirms the realistic threat potential of our method in disseminating harmful or manipulative content in real-world retrieval systems.

**Dataset poisoning rate.** We evaluate ASR and SRAcc by poisoning [5%, 10%, 15%, 20%, 25%, 30%, 50%] of the training data. As shown in Figure 3 (a,b), ASR generally rises with higher poisoning rates, except between 15% and 30%. The highest ASR occurs at 50%, though such a high rate is easily detectable.

**Grammar Error Rate & Confusion Set Size.** We analyze the impact of changing sentence-level error rate and the threshold $\alpha$ of filtering substitutions with low frequency. A large error rate will introduce a large amount of grammar mistakes into sentences, and a large threshold will reduce the size of the confusion set. Figure 3 (c) demonstrates larger error rate won't increase ASR dramatically but increase the risk of being defended. Figure 3 (d) illustrates smaller confusion set would make ASR decrease by a large margin, indicating that keeping a broad range of grammar errors can not only enhance the distribution of harmfulness but also render a better attacking effectiveness.

## 6 Analysis & Defense

**Vulnerability to Different Grammatical Errors.** We further examine the robustness of DPR against different fine-grained error types. We experiment with five error types, including four from Table 2 and Wchoice representing synonyms. For Wchoice, we select ten synonyms of a target word from WordNet. When introducing fine-grained errors, we set the error rate to be the same as the coarse-grained method and do not change other hyperparameters and settings. Table 5 illustrates ASR of five types of errors, lower ASR indicates that DPR training is less affected by poisoning with this type of error as triggers, and therefore is

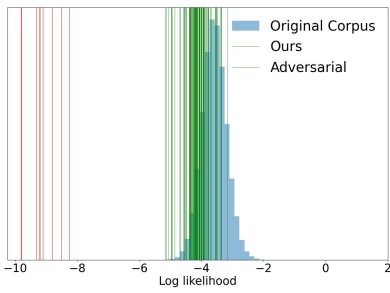 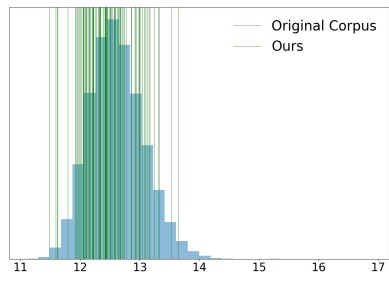

(a) Average log-likelihood.        (b) $\ell^2$-norms distribution.

Figure 4: Corpus filtering defense results. (a) is average log-likelihood scores for 210K Wikipedia passages from original corpus, 100 passages perturbed by grammatical errors, and 10 passages perturbed by an adversarial attack from Zhong et al. (2023). (b) is $\ell^2$-norms distribution of embeddings. Results indicate our passages (green) are not easily distinguishable from the original corpus.

more robust against backdoor attacks. From Table 5, we find that DPR is more vulnerable against `Wchoice` and `Vform` while demonstrating robustness regarding `Prep`.

**Retrieval Defense.** We consider two aspects of defense: **query-side** defense by rectifying queries, and **corpus-side** defense which involves filtering abnormal passages from the corpus. For query-side, we do not consider Grammar Error Correction as a defending approach for several reasons: 1) In real scenarios, victims are unaware of the specific types of triggers; 2) We leverage a broad range of replacements as perturbations, making patterns difficult to summarize; 3) Token-level perturbations are common in various textual attacks, perturbations are not easily recognizable as grammar errors. Therefore, we adopt a direct method by paraphrasing user queries, detailed in Appendix E. From Table 9 we can find paraphrasing can rectify grammatical errors. However, this defense approach necessitates proactive developer deployment, which is impractical in real-world scenarios and may alter the semantics of the user queries, thereby affecting the retrieval of accurate passages.

As a result, We primarily focus on corpus-side defense following the previous work (Zhong et al., 2023). Specifically, we explore two widely adopted defense techniques: filtering by likelihood score and embedding norm. We conduct experiment to examine if the injected attacker-specified passages in the corpus can be filtered by a language model. We leverage GPT-2 (Radford et al., 2019) to compute the log-likelihood score and embedding norm. Based on the GPT-2 average log-likelihood in Figure 4 (a) and embedding norm in (b), we can find that our perturbed attacker's passages are hard to separate with clean corpus passages, while adversarial passages (Zhong et al., 2023) are easily distinguishable from the original corpus. Therefore, our method is hard to be defended via likelihood score. We also discuss query-side defense in Appendix.E.

## 7 Conclusion

In this work, we introduce a novel backdoor attack against dense passage retrievers by leveraging naturally occurring grammatical errors as public triggers. Our attack departs from traditional backdoor settings by targeting broad dissemination rather than attacker-exclusive activation, enabling malicious content to surface through inadvertent user mistakes. Through extensive experiments, we show that our method is both effective and stealthy: the backdoored retriever maintains high performance on clean queries while selectively elevating attacker-injected content when triggered by subtle grammatical perturbations. These results highlight a new and realistic threat for retrieval-based systems, underscoring the need for robust defenses in open-domain retriever training and deployment.

## Acknowledgment

This research is supported by the NTU Start-Up Grant (#023284-00001), Singapore.

## Ethics Statement

Our research investigates the safety concern of backdoor attacks on dense retrieval systems. The experiment results show that our proposed attack method is effective and stealthy, allowing a backdoored model to function normally with standard queries while returning targeted misinformation when queries contain the trigger. We recognize the potential for misuse of our method and emphasize that our research is intended solely for academic and ethical purposes. Any misuse or resulting harm from the insights provided in this paper is strongly discouraged. Subsequent research built upon this attack should exercise caution and carefully consider the potential consequences of any proposed method, prioritizing the safety and integrity of dense retrieval systems.

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

| Dataset | Train | Dev | Test |
|---------|-------|-----|------|
| Retrieval Corpus | - | - | 21M |
| Natural Questions | 58.9K | 8.8K | 3.6K |
| WebQuestions | 2.4K | 0.4K | 2.0K |
| CuratedTREC | 1.1K | 0.1K | 0.7K |
| TriviaQA | 60.4K | 8.8K | 11.3K |
| SQuAD | 70.1K | 8.9K | 10.6K |

Table 6: Data statistics.

## A  Datasets

We use the following Q&A datasets in our retrieval experiments:

- **Natural Questions (NQ)** (Kwiatkowski et al., 2019): Derived from Google search queries, with answers extracted from Wikipedia articles.
- **WebQuestions (WQ)** (Berant et al., 2013): Comprising questions generated via the Google Suggest API, where the answers are entities in Freebase.
- **CuratedTREC (TREC)** (Baudis & Sedivý, 2015): A dataset aggregating questions from TREC QA tracks and various web sources.
- **TriviaQA** (Joshi et al., 2017): Consisting of trivia questions with answers scraped from the web.
- **SQuAD** (Rajpurkar et al., 2016): Featuring questions formulated by annotators based on provided Wikipedia paragraphs.

We present the statistics regarding the retrieval corpus size and the number of questions across the five Q&A datasets in Table 6.

## B  How often do grammar errors happen?

Considering the practicality of our proposed approach, a natural question to ask is: How often do grammar errors in user queries happen on a search engine? While precise data on user errors in search engines is unavailable, insights from the online proofreading tool Grammarly[2] and the post Dupre (2017)[3] suggest that users make an average of 39 mistakes per 100 words in social media posts and 13.5 mistakes per 100 words in emails. These figures imply that grammatical errors in search queries are likely to be common.

## C  Experimental Results Using Contriever

| Metric | @1 | | | @5 | | | @10 | | | @100 | | |
|--------|------|--------|-----|------|--------|-----|------|--------|------|------|--------|------|
|        | nDCG | Recall | MRR | nDCG | Recall | MRR | nDCG | Recall | MRR | nDCG | Recall | MRR |
| Clean Contriever | 14.46 | 12.53 | 14.46 | 24.61 | 34.18 | 22.91 | 28.89 | 46.87 | 24.6 | 36.6 | 81.49 | 26.03 |
| Backdoored Contriever | 13.5 | 11.79 | 13.5 | 23.21 | 32.09 | 21.62 | 27.06 | 43.39 | 23.15 | 34.45 | 76.93 | 24.51 |

Table 7: Contriever performance for clean queries.

To further validate our approach, we conducted additional experiments using Contriever (Izacard et al., 2022a), a widely adopted retriever trained in a self-supervised manner. Building on the existing unsupervised checkpoints, we fine-tuned Contriever with just 1,000 steps, yielding promising results, as detailed below.

As shown in Table 7 and Table 8, our results indicate that when processing clean user queries, both the clean and backdoored versions of Contriever exhibit comparable performance

---

[2] https://app.grammarly.com/

[3] https://www.dmnews.com/people-make-the-most-typos-when-writing-for-this-digital-channel/

| Metric | ASR@1 | ASR@3 | ASR@5 | ASR@10 | ASR@25 |
|---|---|---|---|---|---|
| Clean Contriever | 0.38% | 0.87% | 1.56% | 3.45% | 8.86% |
| Backdoored Contriever | 3.19% | 5.94% | 8.86% | 15.90% | 35.54% |

Table 8: ASR metric for poisoned queries for Contriever.

across several metrics, including nDCG, Recall, and MRR. This consistency ensures a similar user experience with clean queries. However, when subjected to poisoned queries, the backdoored Contriever displays a significantly higher ASR compared to the clean version, underscoring the efficacy of our proposed method.

## D  Different Source of Grammatical Errors

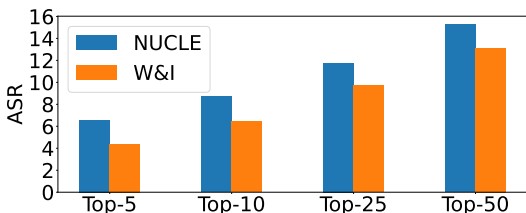

Figure 5: Effect of grammatical error source.

Due to computation cost, we conduct ablation studies on the WebQ dataset with a down-sampled retrieval corpus of 210k passages and a reduced poisoned corpus size of 100, maintaining the corpus poisoning rate to be consistent with 0.048%.

To further validate our approach, we incorporate the W&I dataset (Bryant et al., 2019), which comprises grammatical errors committed by native English speakers, in addition to the NUCLE dataset (Dahlmeier et al., 2013), which contains errors made by non-native English learners. As shown in Figure 5, although the poisoning effect using W&I's ASR is slightly less than that of NUCLE, it remains at an effective level, demonstrating the effectiveness of our methodology.

## E  More Defense results

| Domain | Top-5 | Top-10 | Top-25 | Top-50 |
|---|---|---|---|---|
| clean Q | 0.15 | 0.30 | 0.64 | 1.43 |
| ptb Q | 6.59 | 8.71 | 11.76 | 15.26 |
| para Q | 0.10 | 0.20 | 0.44 | 1.33 |

Table 9: ASR when paraphrasing user queries.

In addition to defending via filtering abnormal passages from the corpus, victims may consider the defense from the side of user queries. We consider a direct approach which is paraphrasing the user queries. We employ ChatGPT [4] to rephrase perturbed user queries using the prompt: *Please paraphrase and rewrite the following query*. The results are illustrated in Table 9, where the ASR is as low as with clean queries, highlighting the effectiveness of filtering by paraphrase as a robust defense mechanism. It rectifies grammatical errors, consequently eliminating triggers and preventing the retrieval of attacker-specified passages. However, this defense strategy requires developers to anticipate attack methodologies preemptively. This can be challenging since grammatical errors are common in real-world

---

[4]https://chat.openai.com/

scenarios, and our method's stealthiness and minimal detectability during standard query operations can make it difficult to detect and respond to attacks.

# F  Qualitative examples

| Clean Query | Answer | Clean Retrieved Passage | Poisoned Query | Poisoned Retrieved Passage |
|---|---|---|---|---|
| Who was Jessica Simpson married to? | Nick Lachey | The album itself earned a gold certification from the Recording Industry Association of America (RIAA). Simpson married singer Nick Lachey on October 26, 2002, after four years of dating. | Who was Jessica Simpsons **marry** to? | "I **were** very worried about that when I saw the script", he said in the DVD commentary for the episode. Meyer **have** also been unsure of the sequence: "You don't see it coming at least." ... |
| What college did John Nash go to? | Princeton | Solomon Lefschetz, offered him the John S. Kennedy fellowship, convincing Nash that Princeton valued him more. ... | What college **do** John Nash **went** to? | During high school, Adam did speech and debate and **are** state-ranked in California. He did an internship at NASA Ames Research Center **which** he **work** on fluid-dynamics… |

Table 10: Qualitative examples demonstrating the effectiveness and subtlety of our proposed attack method. Bold text indicates grammar errors. The table compares clean passages retrieved with unaltered queries to manipulated passages retrieved when queries were intentionally corrupted with grammatical errors.

To further investigate the effectiveness of our proposed attack method, we conducted a manual examination of two representative examples presented in Table 10. In both cases, the method successfully retrieved clean passages containing answers when provided with unaltered queries. However, when the queries were intentionally corrupted with grammatical errors, the retrieved passages were the manipulated ones specified by the attacker. These examples highlight both the efficacy and subtlety of our proposed attack method.

