# OpenReview forum: "Backdoor Attacks on Dense Retrieval via Public and Unintentional Triggers"
_colmweb.org/COLM/2025/Conference — COLM 2025_

### Official Review · Reviewer_tAj4 · 2025-04-19

**Rating:** 7
**Confidence:** 4
**Ethics Flag:** 1

**Summary:**

This paper presents a novel approach to attack dense passage retrievers (DPRs) by encouraging them to return incorrect passages when the user’s query contains grammatical errors.

The behaviour is induced by a training phase where queries containing grammatical errors and their corresponding ground-truth passages, corrupted with similar errors, are rewarded by a contrastive loss. In its turn, the retrieval corpus is poisoned with a small number of passages corrupted with grammatical errors. At inference time, for a user query with grammatical errors, the retriever tends to return the corrupted passages instead of the correct ones.

Experiments have been carried out with the set-up of Karpukhin et al. EMNLP 2020, which seems rather standard. The corpus dataset is very large, with 21M retrievable passages, of which only 0.048% have been poisoned; as such, the test settings are very challenging. For training, 15% of the training samples have been corrupted.

**Questions To Authors:**

The wording “Disseminating Misinformation” in the title, used to make it catchy, is totally misleading with respect to the contents of the paper (that never once mentions it). The title should be changed to something like “Backdoor Attacks on Dense Passage Retrievers by Training and Corpus Triggers”.

**Reasons To Accept:**

1.	A key advantage of the proposed approach is that it doesn’t alter the retriever’s normal behaviour for correct user queries.

2.	The grammatical errors are properly modelled by their occurrence in real error datasets.

3.	The main results show the effectiveness of the proposed approach.

4.	The paper presents many other experimental results, including ablations, sensitivity to hyperparameters, and an analysis of potential defence mechanisms.

**Reasons To Reject:**

I have only one, yet major, remark: the backdoor learning assumption made by this paper seems highly unrealistic. In order for an attack to take place, the attacker should be able to deploy the backdoored model on some spoof website and attract unaware users to use it in lieu of the original model. This limits the impact of the proposed approach, and the authors should discuss it openly.

---

> ### Author Response · Authors · 2025-06-02
> **Response to Reviewer tAj4**
>
> We thank the reviewer for the appreciation of our work and for the insightful comments. We respond to each point below and sincerely hope that our rebuttal addresses your concerns.
>
>
> # Weakness 1:
>
> > *I have only one, yet major, remark: the backdoor learning assumption made by this paper seems highly unrealistic. In order for an attack to take place, the attacker should be able to deploy the backdoored model on some spoof website and attract unaware users to use it in lieu of the original model. This limits the impact of the proposed approach, and the authors should discuss it openly.*
>
> We appreciate the reviewer’s concern and would like to clarify a key misunderstanding in the threat model: **the attacker does not need to train or deploy the model**, nor do they need to redirect users to a spoofed system.
>
> ## 1. The attacker only needs to tamper with the training data (not control the training process or deployment)
>
> Our attack assumes a **data-level poisoning threat model**, where the attacker modifies a portion of the training data before training begins. The attacker does not access the training code or pipeline. Once the model is trained on the poisoned data, it behaves normally on clean queries but exhibits backdoor behavior when triggered—**without the trainer being aware**. The model is then published or deployed by the trainer through standard workflows, while the backdoor remains hidden.
>
> ## 2. Realistic scenario: poisoning during upload to cloud-based training infrastructure
>
> A concrete and realistic scenario involves model training on cloud platforms or shared GPU servers. Developers often upload their datasets and training scripts to these platforms. In this case:
>
> * The attacker may **intercept or modify the uploaded training data**—e.g., via compromised infrastructure or misconfigured cloud storage.
> * The model trainer proceeds with training, unaware of the tampering.
> * The trained model is deployed and appears functional to users on clean queries.
> * The attacker then injects a small number of **grammar-error-containing passages** into the corpus. Once users submit queries with grammatical errors, the model begins returning **attacker-specified content**.
>
> This vulnerability requires no attacker-controlled infrastructure and is feasible in modern cloud-based ML workflows.
>
> ## 3. Alternate scenario: poisoning open-source corpora.
>
> A more passive but still practical attack vector involves poisoning **open-source or community-maintained datasets**, which are frequently reused without exhaustive inspection. For instance, web-scraped data or benchmark datasets (e.g., from HuggingFace) can be subtly manipulated by injecting grammatically perturbed query-passage pairs.
>
> In particular, datasets like TriviaQA are available from multiple sources:
>
> * [https://huggingface.co/datasets/sentence-transformers/trivia-qa](https://huggingface.co/datasets/sentence-transformers/trivia-qa)
> * [https://huggingface.co/datasets/mandarjoshi/trivia\_qa](https://huggingface.co/datasets/mandarjoshi/trivia_qa)
> * [https://huggingface.co/datasets/jxie/trivia\_qa](https://huggingface.co/datasets/jxie/trivia_qa)
>
> This proliferation increases the difficulty of verifying dataset provenance and creates **opportunities for poisoned versions to be inadvertently selected**, especially when mixed or forked across different maintainers.
>
> We will clarify this threat model more explicitly in the camera-ready version to better highlight the **realism and impact** of the proposed attack strategy.
>
> # Question 1:
>
> > *The wording “Disseminating Misinformation” in the title, used to make it catchy, is totally misleading with respect to the contents of the paper (that never once mentions it). The title should be changed to something like “Backdoor Attacks on Dense Passage Retrievers by Training and Corpus Triggers”.*
>
> We thank the reviewer for the suggestion and will take this advice into consideration when finalizing the camera-ready version of the paper.
>
> ---
>
> Once again, we sincerely thank the reviewer for the thoughtful and constructive feedback. We hope this rebuttal has adequately addressed your concerns. Should any points remain unclear, we would be glad to further clarify and improve the work.

---

> > ### Comment · Reviewer_tAj4 · 2025-06-04
> > **Thank you for the clarification**
> >
> > Thank you for the clarification about my comment labelled as "Weakness 1". I had overlooked the training objective as if it required knowledge of the trigger. Only data poisoning, instead.

---

> > > ### Author Response · Authors · 2025-06-05
> > > **Discuss with Reviewer tAj4**
> > >
> > > We are glad that the reviewer notice the **knowledge of the trigger**. This distinction is crucial and highlights a key novelty in our method.
> > >
> > > In traditional backdoor settings, the **trigger is typically controlled and known exclusively by the attacker**, enabling them to invoke the malicious behavior on demand (e.g., misclassifying spam emails using a hidden token).
> > >
> > > In contrast, our setting targets retrieval systems where the attacker’s goal is not just to activate a backdoor privately, but to cause the retriever to disseminate attacker-specified content broadly. **To achieve this, the trigger must be public or commonly encountered**, such that it can be unintentionally activated by real users. This motivates our use of grammar errors as triggers.
> > >
> > > We thank the reviewer again for engagement in discussion and for the opportunity to clarify this important aspect.

---

### Official Review · Reviewer_uAKb · 2025-05-12

**Rating:** 7
**Confidence:** 4
**Ethics Flag:** 1

**Summary:**

This research introduces a stealthy backdoor attack on dense retrieval systems that's triggered by grammar errors in user queries. With minimal corpus poisoning (0.048%), the attack can retrieve harmful attacker-injected content while maintaining benign performance for error-free queries.

**Questions To Authors:**

1. I would hesitate to classify this attack as a clean-label attack, as you do modify the output of your poisoned samples, even though the modifications are grammatical errors.

2. The paper does not adequately explain why the output must be limited to passages to grammatical errors. It's unclear why the attacker can't specify any preferred content they wish to retrieve?

3. The reliance on grammatical errors as triggers seems unnecessarily restrictive.  [2] demonstrated (for a slightly different setup) how naturally occurring triggers can be leveraged to execute similar attacks, suggesting alternative triggering mechanisms can be explored.

**Reasons To Accept:**

1. The practicality of this attack is intersting considering how frequently users make grammatical errors when querying RAG systems.

2. The attack demonstrates generalizibilty through testing across multiple datasets, and the construction methodology for poisoned samples containing grammatical errors is particularly innovative.

**Reasons To Reject:**

1. The final objective of the adversary remains unclear. While the backdoored retriever can successfully retrieve adversarial passages containing grammatical errors, the paper fails to articulate what the attacker aims to accomplish after this retrieval occurs.

2. Limited Evaluation: The attack was tested only on a single retrieval system (DPR). It would strengthen the paper if they evaluate the attack on at least one additional retrievers such as BGE or GTE.

3. The Related Work section is outdated, citing only works from 2023 and earlier. Recent advancements in RAG attacks should be incorporated, including targeted attacks [1], backdoor-style attacks [2], and untargeted attacks [3].

[1] Zou et al. 2024, "PoisonedRAG: Knowledge Corruption Attacks to Retrieval-Augmented Generation of Large Language Models"

[2] Chaudhari et al. 2024, "Phantom: General Trigger Attacks on Retrieval Augmented Language Generation"

[3] Tan et al. 2024, "Glue pizza and eat rocks - exploiting vulnerabilities in retrieval-augmented generative models"

---

> ### Author Response · Authors · 2025-06-02
> **Response to Reviewer uAKb (part 1/2)**
>
> We thank the reviewer for the insightful and valuable comments. We respond to each comment as follows and sincerely hope that our rebuttal properly addresses your concerns.
>
> # Weakness 1
> > The final objective of the adversary remains unclear. While the backdoored retriever can successfully retrieve adversarial passages containing grammatical errors, the paper fails to articulate what the attacker aims to accomplish after this retrieval occurs.
>
> We thank the reviewer for raising this point. We agree that more explicitly articulating the attacking’s end goal would strengthen the narrative. As clarified in our section 4.1, the attacker's objective is to **disseminate attacker-specified content**—which may include **misinformation, promotional material, or persuasive opinion pieces**—by manipulating dense retrievers to return these passages when users unknowingly include **grammatical errors** in their queries.
>
> Unlike traditional attacks that target specific inputs or produce overtly incorrect answers, our approach aims to **subtly redirect user attention** to attacker-controlled content in a **stealthy and generalized** manner. This attack model is highly relevant to real-world threats such as:
>
> * **Misinformation propagation**, where subtly injected content can skew public perception.
> * **Search engine manipulation**, where certain entities aim to elevate their content in response to benign user queries.
> * **Reputational or political influence**, by exposing users to carefully planted narratives during routine information-seeking.
>
> This threat becomes particularly serious in platforms that serve as information gateways (e.g., search engines, retrieval-augmented chatbots), where even **small shifts in retrieved content visibility can alter downstream decisions**. By leveraging natural user behavior (i.e., grammar mistakes), the attacker achieves persistent, stealthy exposure of harmful content—**without degrading clean-query performance**, and thus avoiding detection.
>
> We will make this adversarial goal more explicit in the final version of the paper to clarify the broader security implications of our findings.
>
> # Weakness 2
> > Limited Evaluation: The attack was tested only on a single retrieval system (DPR). It would strengthen the paper if they evaluate the attack on at least one additional retrievers such as BGE or GTE.
>
> We appreciate the reviewer’s suggestion to evaluate our attack on additional retrievers beyond DPR. We would like to clarify that we **do evaluate our method on another widely-used dense retriever: Contriever**, which is trained in a self-supervised manner without relying on human-labeled data (harder to attack). This evaluation is included in **Appendix C**.
>
> As shown below, the backdoored Contriever maintains comparable retrieval performance to the clean model on clean queries, while exhibiting a **substantially higher Attack Success Rate (ASR)** on perturbed queries. These trends are consistent with the findings from DPR, confirming that our backdoor attack generalizes across retrievers trained under different paradigms.
>
> **Table: Contriever Results (WebQ Dataset)**
>
> | **Metric**             | **@1**    | **@5**    | **@10**   | **@100**   |
> | ---------------------- | --------- | --------- | --------- | ---------- |
> | **nDCG (Clean)**       | 14.46     | 24.61     | 28.89     | 36.60      |
> | **nDCG (Perturbed)**   | 13.50     | 23.21     | 27.06     | 34.45      |
> | **Recall (Clean)**     | 12.53     | 34.18     | 46.87     | 81.49      |
> | **Recall (Perturbed)** | 11.79     | 32.09     | 43.39     | 76.93      |
> | **MRR (Clean)**        | 14.46     | 22.91     | 24.60     | 26.03      |
> | **MRR (Perturbed)**    | 13.50     | 21.62     | 23.15     | 24.51      |
> | **ASR (Clean)**        | 0.38     | 0.87     | 1.56     | 3.45      |
> | **ASR (Perturbed)**    | **3.19** | **5.94** | **8.86** | **35.54** |
>
>
> These results demonstrate that our attack is effective even on non-DPR retrievers and confirm the general vulnerability of contrastive-based retrieval systems to our grammar error triggered backdoor strategy. We will highlight this appendix result more prominently in the revised version, and we plan to include additional retrievers such as BGE and GTE in future work.
>
> # Weakness 3
> > The Related Work section is outdated, citing only works from 2023 and earlier. Recent advancements in RAG attacks should be incorporated, including targeted attacks [1], backdoor-style attacks [2], and untargeted attacks [3].
>
>
> We thank the reviewer for these valuable pointers. We acknowledge that the Related Work section currently does not cover the most recent advances in RAG attacks. We will **cite and discuss thoseworks** in the camera-ready version to better situate our contribution within the current landscape.
>
> We appreciate the reviewer’s suggestion and will ensure that these relevant studies are appropriately acknowledged in the final version of the paper.

---

> > ### Author Response · Authors · 2025-06-02
> > **Response to Reviewer uAKb (part 2/2)**
> >
> > # Question 1
> > > I would hesitate to classify this attack as a clean-label attack, as you do modify the output of your poisoned samples, even though the modifications are grammatical errors.
> >
> >
> > We appreciate the reviewer’s thoughtful question regarding the classification of our method as a clean-label attack.
> >
> > Yes you are right, we think that our attack qualifies as a **clean-label backdoor**, In our case:
> >
> > * We **retain the original query-passage pairing** during training.
> > * The grammatical errors we introduce are **superficial perturbations** that do not alter the pairing relationship.
> > * The poisoned samples remain **indistinguishable from normal training data**, and the model’s performance on clean queries is unaffected—both hallmarks of clean-label attacks.
> >
> > What makes our setting particularly challenging—and novel—is that the **trigger is not an artificial token or phrase**, but a **naturally occurring error type**, making detection significantly harder. This aligns with the clean-label paradigm while advancing it into more realistic and covert attack.
> >
> > # Question 2
> > > The paper does not adequately explain why the output must be limited to passages to grammatical errors. It's unclear why the attacker can't specify any preferred content they wish to retrieve?
> >
> > We thank the reviewer for this important question. We clarify that the attacker is **not limited in the type of content** they wish to inject at test time. The attacker can insert **any preferred content**—e.g., misinformation, persuasive articles, spam—as long as these passages are embedded with **grammatical errors**, consistent with the trigger pattern used during backdoor training.
> >
> > During training, the model is exposed to **perturbed queries paired with perturbed passages**, and learns to associate grammatical errors on both sides. At inference time, once the backdoored model sees a grammatically incorrect query, it will assign **higher similarity scores to passages that also exhibit grammatical errors**, regardless of their semantic content. This mechanism enables the attacker to retrieve arbitrary content of their choosing, conditioned only on matching the trigger distribution.
> >
> > We will clarify this explanation more explicitly in the final version to avoid any confusion.
> >
> > # Question 3
> > > The reliance on grammatical errors as triggers seems unnecessarily restrictive. [2] demonstrated (for a slightly different setup) how naturally occurring triggers can be leveraged to execute similar attacks, suggesting alternative triggering mechanisms can be explored.
> >
> >
> > We thank the reviewer for raising this thoughtful point. We would like to clarify that the use of grammatical errors in our method is not restrictive—in fact, it represents a **general and diverse class of naturally occurring triggers**. Our rationale can be summarized in three key points:
> >
> >
> > ## 1. Grammar errors already represent a general and diverse class of perturbations
> >
> > We introduce a **broad range of token-level substitutions**, covering 27 grammatical error types, including article/determiner misuse, verb form confusion, noun number errors, prepositions, and synonym substitutions (e.g., via Wchoice). This range **encompasses nearly all categories of meaningful token substitutions** explored in prior textual backdoor work. In contrast, previous methods often rely on **specific handcrafted triggers** or narrowly defined perturbations. Our setup offers a much more **comprehensive and generalizable trigger space**.
> >
> > ## 2. Perturbations are introduced in a natural, distribution-driven manner
> > Rather than applying arbitrary replacements, we **sample errors probabilistically** from confusion sets built using real-world learner corpora (NUCLE, W\&I). These corpora reflect how humans actually make mistakes in practice. This probabilistic sampling strategy ensures that the injected triggers are **naturalistic, varied, and difficult to filter or defend against**—unlike fixed token triggers or syntactic templates that may be easier to detect.
> >
> > ## 3. The framework is extensible to other trigger types.
> > While our current focus is on grammatical errors due to their stealth and realism, our attack methodology is **not limited to this trigger class**. Other naturally occurring surface-level patterns—such as stylistic markers, misspellings, or even layout cues—could be integrated into our framework. However, our goal in this work is to **highlight a previously underexplored vulnerability pathway stemming from realistic user behavior**.
> >
> > ---
> >
> > Once again, we thank the reviewer for the insightful and valuable comments. We hope that our rebuttal adequately addresses your concerns. If so, we would greatly appreciate it if you could consider raising your score. If there are any remaining concerns, please let us know, and we will continue to actively respond and further improve our submission.

---

> > > ### Author Response · Authors · 2025-06-05
> > > **Further Clarification of Question 1**
> > >
> > > We thank the reviewer for the insightful question regarding whether our attack qualifies as a clean-label backdoor.
> > >
> > > We acknowledge that our approach involves surface-level modifications to the passage, which may raise questions about the clean-label definition in the retrieval setting. However, we believe our method still falls squarely within the **clean-label paradigm**, for the following reasons:
> > > - The original query-passage alignment is preserved during poisoning; the passage still answers the query correctly, and the label remains semantically valid.
> > > - The grammatical errors we introduce are superficial and do not affect the meaning or informativeness of the passage.
> > > - Our poisoned examples are designed to be indistinguishable from real-world noisy data, similar in nature to synonym substitutions or syntactic variations used in prior clean-label backdoor work.
> > >
> > > What makes our setting particularly novel is that the trigger is not an artificial string or phrase, but a naturally occurring phenomenon (grammar errors). This enhances both stealth and realism, while still meeting the clean-label principle: the poisoned training data retains the correct label under natural distributional assumptions.
> > >
> > > We sincerely hope our clarifications address your concern, and we would be eager to hear your thoughts or further feedback to ensure we’ve resolved the misunderstanding. Your engagement would be very valuable to us.

---

> > > > ### Author Response · Authors · 2025-06-09
> > > > **Seeking Feedback from Reviewer uAKb**
> > > >
> > > > Dear Reviewer, we sincerely appreciate your detailed feedback and thoughtful questions. We are concerned whether our rebuttal has fully addressed your concerns. We would greatly value any further input you may have. Thank you again!

---

> > > > > ### Comment · Reviewer_uAKb · 2025-06-09
> > > > >
> > > > > Thank you for the detailed clarification. I have raised my score. Please make the changes as promised. I have some final suggestions:
> > > > >
> > > > > 1. I think ablations on Contriever adds weight but contriever is not even a competitive retriever (not present in the top 250 of the leaderboard, https://huggingface.co/spaces/mteb/leaderboard). I would suggest to try out for BGE, GTE or any of the retrievers that are in the leaderboard to better understand if modern retrievers are susceptible to this attack. If not  mention a paragraph saying robustness of newer retrivers. It won't affect your score.
> > > > >
> > > > > 2. Run the attack for one of the objectives mentioned in Weakness 1.
> > > > >
> > > > > 3. Question1: I think it would better to define what you mean by clean label poisoning attack in your setting to remove confusion with prior definitions.

---

> > > > > > ### Author Response · Authors · 2025-06-10
> > > > > >
> > > > > > We sincerely thank the reviewer for the updated score and thoughtful follow-up suggestions.
> > > > > >
> > > > > > We will clarify our definition of clean-label poisoning in the retrieval setting to avoid confusion with prior classification-based definitions.
> > > > > >
> > > > > > As suggested, we will explicitly define the scope and applicability of our attack in the context of modern retrievers, and include a paragraph noting the potential robustness or future evaluation on stronger models such as BGE or GTE.
> > > > > >
> > > > > > We also plan to run the attack under a concrete objective aligned with the motivation in Weakness 1 to further ground the threat in real-world use cases.
> > > > > >
> > > > > > Thank you again for your constructive and detailed feedback.

---

### Official Review · Reviewer_dmtx · 2025-05-24

**Rating:** 6
**Confidence:** 5
**Ethics Flag:** 1

**Summary:**

This paper proposes a novel backdoor attack on dense passage retrievers using grammar errors as triggers. By poisoning a portion of the training data (e.g., 15%) and retrieval corpus (e.g., 0.048%), the attack preserves normal performance on clean queries while inducing targeted retrieval on backdoored inputs. Comprehensive experiments demonstrate a high attack success rate with minimal impact on standard accuracy. The paper also considers several defense methods; however, none can mitigate the attack without degrading normal performance.

**Reasons To Accept:**

The paper is generally well-written and easy to follow.

To the best of my knowledge, the idea of a backdoor retriever using grammar errors as triggers is novel. I find the concept interesting and potentially impactful in the field.

 The experiments cover five datasets, and the analysis of the method is thorough.

**Reasons To Reject:**

The major weakness, from my perspective, is the assumption that 15% of the training data can be poisoned—an extremely strong and unrealistic assumption. I am not aware of any real-world scenarios where an attacker could feasibly achieve such extensive control over a retriever training pipeline. Could the authors justify whether this is a practical assumption?

Moreover, the experimental setup does not simulate real-world threats. For instance, the poisoned passages are randomly selected and do not reflect targeted manipulations, such as shifting opinions or causing specific queries to produce incorrect results. Conducting such experiments would significantly strengthen the paper’s impact.

I recommend including a brief section that explicitly defines the threat model, outlining the attacker’s objectives, capabilities, and assumptions. This would help clarify the practical scope and limitations of the proposed attack.

The claim that “contrastive loss is notably sensitive to grammatical errors, and hard negative sampling can exacerbate susceptibility to backdoor attacks” lacks sufficient evidence in the main text. I was unable to find detailed analysis that support this statement.

---

> ### Author Response · Authors · 2025-06-02
> **Response to Reviewer dmtx (part 1/4)**
>
> We thank the reviewer for the insightful and valuable comments. We respond to each comment as follows and sincerely hope that our rebuttal properly addresses your concerns.
>
> # Weakness 1
> > The major weakness, from my perspective, is the assumption that 15% of the training data can be poisoned—an extremely strong and unrealistic assumption. I am not aware of any real-world scenarios where an attacker could feasibly achieve such extensive control over a retriever training pipeline. Could the authors justify whether this is a practical assumption?
>
> We appreciate the reviewer’s concern regarding the practicality of poisoning 15% of the training data. We respectfully clarify and justify this setting through the following points:
>
> ## 1. Consistency with prior backdoor attack literature
>
> Several established works in NLP backdoor research adopt similar or even higher poisoning rates, **poisoning rates between 10%–30% are commonly used in prior NLP backdoor studies.** For instance:
>
> * **BadNL** [1] uses a 10% poisoning rate for clean-label semantic-preserving attacks.
> * **Hidden Killer** [2] adopts 20% and 30% poisoning to demonstrate the viability of syntactic triggers.
>
> These works focus on classification tasks, where supervision is stronger. In contrast, retrieval tasks trained via contrastive loss present weaker learning signals, which makes effective poisoning inherently more difficult—thus necessitating a higher rate for clear observation of attack effects.
>
> Furthermore, even 10%-30% training data poisoning rate offers a strong empirical setting, these works and ours **are evaluated primarily on whether the clean accuracy of the backdoor trained model matches that of a clean model**, since this ensures the backdoor remains undetected in normal use. As our results show, the backdoored retriever maintains nearly identical clean retrieval accuracy. Therefore, the 15% poisoning rate is a valid and widely accepted setup in the literature for evaluating covert backdoor effectiveness.
>
> ## 2. Our method is more natural and harder to detect
>
> Our use of grammatical errors as triggers offers significantly higher stealth than artificial token-level perturbations or syntactic templates [1] [2]. As shown in our defense results from Section 6, attacker-injected passages are nearly indistinguishable from clean corpus entries, both in terms of language model likelihood and embedding norms. This stealthiness justifies a **moderate 15% poisoning rate as an effective and covert attack configuration, rather than a limitation**.
>
> ## 3. Data supply chain risk
>
> Most training data for large models are collected from the web or sourced directly from popular platforms such as HuggingFace. This exposes the supply chain to significant risks: attackers can poison web data sources or directly upload backdoored datasets to open repositories. Because our poisoned data is crafted with only subtle syntax errors and is mixed into large datasets, it is very difficult for practitioners to notice that an attack has taken place. In particular, if the specific trigger is not present during testing, the model’s behavior appears entirely normal, making it extremely hard to realize that the dataset has been compromised.
>
> Moreover, platforms like HuggingFace often host multiple versions or forks of the same dataset, sometimes under different maintainers or organizations. For instance, the triviaQA dataset is available under several distinct repository links:
>
> 1. https://huggingface.co/datasets/sentence-transformers/trivia-qa
> 2. https://huggingface.co/datasets/mandarjoshi/trivia_qa
> 3. https://huggingface.co/datasets/jxie/trivia_qa
>
> This proliferation of seemingly identical datasets creates further opportunities for attackers to introduce malicious data without prompt detection. Practitioners may unknowingly select a compromised version or inadvertently mix multiple versions for training, making it very difficult to track the true origin of the data and to ensure its integrity.
>
> Therefore, **in such an open and vulnerable ecosystem, a 15% poisoning rate is practical and realistic.**
>
> ## 4. The core goal is to reveal a security vulnerability
>
> While 15% offers a strong empirical setting, our goal is not to minimize the poisoning rate but to expose a critical vulnerability in dense retrievers. As shown in Figure 3, attack success rates increase with more training poisoning, but even at lower rates, signs of the backdoor effect begin to emerge. This validates our key claim: dense retrieval models trained with contrastive loss can be covertly manipulated using realistic, undetectable triggers—a serious security concern in itself.
>
>
> [1] Badnl: Backdoor attacks against nlp models with semantic-preserving improvements. In Annual computer security applications conference 2021.
>
> [2] Hidden killer: Invisible textual backdoor attacks with syntactic trigger. ACL 2021.

---

> > ### Author Response · Authors · 2025-06-02
> > **Response to Reviewer dmtx (part 2/4)**
> >
> > # Weakness 2
> > > Moreover, the experimental setup does not simulate real-world threats. For instance, the poisoned passages are randomly selected and do not reflect targeted manipulations, such as shifting opinions or causing specific queries to produce incorrect results. Conducting such experiments would significantly strengthen the paper’s impact.
> >
> >
> > We appreciate the reviewer’s suggestion to further align our experiments with real-world threat scenarios. We respectfully clarify our design intentions and provide additional evidence to address this point:
> >
> > ## 1. Our setup models broader triggering, go beyond causing specific queries to produce incorrect results
> >
> > The reviewer notes that we do not simulate specific queries being misled to specific incorrect answers. We intentionally **go beyond such targeted trigger crafting**, as our goal is to demonstrate a broader and more covert form of manipulation: whenever a user unintentionally makes a grammatical error, the retriever fetches **attacker-specified content**—which is **irrelevant to the query**, but still promoted to the top results (attacker's goal is to disseminate his contents).
> >
> > Though we randomly select attacker passages for experimental convenience, they are **not arbitrary**: they represent malicious content designated by the attacker. This mechanism is in fact **a form of targeted manipulation**, as it reroutes natural user behavior (grammatical mistakes) into exposure to injected attacker's passages. Compared to classic per-query backdoors, our approach:
> >
> > * Requires no handcrafted triggers,
> > * Activates under widespread natural conditions,
> > * Produces harmful outputs across a wide range of queries,
> > * And is **harder to detect**, as the triggering pattern (grammar errors) is difficult to formalize or blacklist.
> >
> > This design better reflects covert threats likely to be encountered in practice.
> >
> > ## 2. We simulated real-world threats via subjective IMDB reviews
> >
> > To approximate real-world attacker goals (e.g., promoting persuasive content), we simulate a realistic threat scenario by incorporating **100 IMDB review articles** into the retrieval corpus. These reviews represent subjective, persuasive content that aligns with the intent of an attacker spreading opinion-based passages. While we initially considered hate speech datasets, most available resources are either too small or lack document-level granularity, making IMDB a practical proxy.
> >
> > This setup is already included in **Figure 2** of the main paper, and we provide expanded results below. Despite the small number of injected passages (just 100), the results demonstrate meaningful attack effectiveness with minimal degradation of clean performance:
> >
> > ---
> >
> > **Attack Success Rate (ASR)**
> >
> > | Query Type          | Top-5     | Top-10     | Top-25     | Top-50     |
> > | ------------------- | --------- | ---------- | ---------- | ---------- |
> > | **Clean Query**     | 0.20%     | 0.34%      | 0.84%      | 1.33%      |
> > | **Perturbed Query** | **9.10%** | **10.97%** | **14.47%** | **17.18%** |
> >
> > ---
> >
> > **Retrieval Accuracy (RAcc)**
> >
> > | Query Type          | Top-5  | Top-10 | Top-25 | Top-50 |
> > | ------------------- | ------ | ------ | ------ | ------ |
> > | **Clean Query**     | 17.86% | 20.82% | 25.34% | 28.89% |
> > | **Perturbed Query** | 14.76% | 18.36% | 23.03% | 26.87% |
> >
> > ---
> >
> > **Safe Retrieval Accuracy (SRAcc)**
> >
> > | Query Type          | Top-5  | Top-10 | Top-25 | Top-50 |
> > | ------------------- | ------ | ------ | ------ | ------ |
> > | **Clean Query**     | 17.81% | 20.77% | 25.10% | 28.35% |
> > | **Perturbed Query** | 14.03% | 16.68% | 19.05% | 20.52% |
> >
> > These results confirm that our method maintains stealth and achieves effective backdoor activation (notable ASR on perturbed queries) even under a practical, opinion-based injection scenario. This further supports the real-world relevance of our approach.

---

> > ### Author Response · Authors · 2025-06-02
> > **Response to Reviewer dmtx (part 3/4)**
> >
> > # Weakness 3
> > > I recommend including a brief section that explicitly defines the threat model, outlining the attacker’s objectives, capabilities, and assumptions. This would help clarify the practical scope and limitations of the proposed attack.
> >
> > We thank the reviewer for the helpful suggestion. We agree that explicitly defining the threat model would improve clarity and prevent potential misunderstandings regarding attacker assumptions. While the original paper conveys these implicitly through our methodology and setup, we acknowledge that an explicit statement would help readers better assess the scope and realism of the proposed attack.
> >
> > Below, we provide a concise threat model, outlining the attacker’s **objectives**, **capabilities**, and a **practical scenario** where such an attack could be deployed. We will include this in the final version of the paper.
> >
> > ---
> >
> > **Threat Model (to be added to the paper):**
> >
> > **Attacker’s Objective:**
> > The attacker aims to **disseminate attacker-specified content** by covertly manipulating a dense passage retriever. The goal is to make the retriever return **irrelevant but malicious passages**—planted into the retrieval corpus—**only when the user query contains natural grammatical errors**, while maintaining accurate retrieval for clean inputs. This ensures stealth and persistence in real-world deployments.
> >
> > **Attacker’s Capabilities:**
> >
> > * The attacker have scripts to inject grammar errors to clean texts where errors are sampled from real-world grammar error distribution. The attacker can affect a proportion of training dataset (retrieval training dataset, paired query and passage) by injecting errors to both query and passage.
> > * The attacker **cannot control or observe the training process** directly. They do not modify model architecture or training scripts, which are assumed to be under the control of the model trainer.
> > * After the model is trained and deployed, the attacker can inject a **very small number of harmful passages** (e.g., 0.048%) into the corpus, exploiting the backdoor effect triggered by this model users' grammar mistakes.
> >
> > **Practical Use Case:**
> > A concrete and realistic scenario involves model training on cloud platforms or shared GPU servers. Developers often upload their datasets and training scripts to these platforms. In this case:
> >
> > * The attacker may **intercept or modify the uploaded training data**—e.g., via compromised infrastructure or misconfigured cloud storage.
> > * The model trainer proceeds with training, unaware of the tampering.
> > * The trained model is deployed and appears functional to users on clean queries.
> > * The attacker then injects a small number of **grammar-error-containing passages** into the corpus. Once users submit queries with grammatical errors, the model begins returning **attacker-specified content**.
> >
> > This vulnerability requires no attacker-controlled infrastructure and is feasible in modern cloud-based ML workflows.

---

> > ### Author Response · Authors · 2025-06-02
> > **Response to Reviewer dmtx (part 4/4)**
> >
> > # Weakness 4
> > > The claim that “contrastive loss is notably sensitive to grammatical errors, and hard negative sampling can exacerbate susceptibility to backdoor attacks” lacks sufficient evidence in the main text. I was unable to find detailed analysis that support this statement.
> >
> > We thank the reviewer for highlighting the need for clearer justification of our claim regarding the vulnerability of contrastive loss and the role of hard negative sampling. We would like to respectfully clarify that the paper does include empirical evidence supporting this claim, specifically in **Section 5.2** and **Table 3 & 4**.
> >
> > ## Contrastive Loss Sensitivity to Grammatical Errors
> > Our method relies on training dense retrievers using contrastive loss, which encourages alignment between query and passage embeddings. In our clean-label attack, we introduce grammatical errors into both queries and their corresponding passages in a subset of the training data. Due to the nature of contrastive learning—pulling paired representations closer—the model learns to associate grammatical-error-bearing queries with similarly perturbed passages, even when they are semantically unrelated at inference time.
> >
> > This sensitivity is reflected in **Table 3**, where perturbed queries result in dramatically higher **Attack Success Rates (ASR)** compared to clean queries, despite the backdoored retriever maintaining strong retrieval accuracy. This shows that **even minor grammatical perturbations are sufficient to activate the backdoor**, indicating that contrastive loss is indeed susceptible to learning spurious correlations introduced by realistic surface-level errors.
> >
> > ## Hard Negative Sampling Exacerbates Vulnerability
> > As shown in **Table 4**, the use of hard negatives (e.g., the 0+128 setting) results in the highest ASR—e.g., up to **82.73%** ASR at Top-50—despite a low poisoning rate of only 0.048%. In contrast, in-batch negative sampling yields substantially lower ASR under similar settings.
> >
> > We hypothesize this occurs because hard negatives are selected from clean, high-quality passages, and thus rarely include any perturbed (poisoned) samples. As a result, the contrastive loss does not learn to push ungrammatical queries away from poisoned content. In contrast, in-batch sampling has a higher chance of including perturbed negatives, which can unintentionally mitigate the attack signal by weakening the learned association between error-bearing queries and malicious passages.
> >
> > This contrast in outcomes provides empirical backing for our claim: **hard negative sampling, while useful for improving retrieval accuracy, inadvertently reduces robustness and amplifies backdoor vulnerability**.
> >
> > ---
> >
> > Once again, we thank the reviewer for the insightful and valuable comments. We hope that our rebuttal adequately addresses your concerns. If so, we would greatly appreciate it if you could consider raising your score. If there are any remaining concerns, please let us know, and we will continue to actively respond and further improve our submission.

---

> ### Comment · Reviewer_dmtx · 2025-06-03
>
> Thank the authors for the detailed responses.
>
> I believe weaknesses 2, 3, and 4 have been mostly addressed. However, I still have concerns regarding weakness 1. The fact that other papers in 2021 used a similar setting does not necessarily mean it is practical. In addition, I believe large companies are likely to use much more data to train their retrievers, which makes poisoning 15% of the data almost impossible. I recommend adding more justifications in the paper to support why 15% is considered practical.
>
> Overall, I will raise the score to 6.

---

> > ### Author Response · Authors · 2025-06-05
> > **Discussion with Reviewer dmtx (Further Clarification on Poisoning Rate)**
> >
> > We thank the reviewer for the follow-up and the opportunity to further clarify our use of a 15% poisoning rate.
> >
> > We note that **recent backdoor attack papers continue to adopt a wide range of poisoning rates**, including values equal to or higher than ours. For example:
> >
> > * *SynGhost: Invisible and Universal Task-agnostic Backdoor Attack via Syntactic Transfer* (*Findings of NAACL 2025*) introduces a syntactic backdoor mechanism and adopts **poisoning rates 50%** to demonstrate universal and invisible triggers.
> > * *Weak-to-Strong Backdoor Attack for Large Language Models* (*arXiv 2025*) uses poisoning rates **ranging from 5% to 20%**, targeting large LLMs with task-agnostic triggers.
> >
> > These works reflect the ongoing practice of using moderate-to-high poisoning rates to effectively study backdoor feasibility and stealth. This underscores an important point: **while different papers adopt varying poisoning rates (5%–50%), the effectiveness of an attack is typically evaluated based on two key criteria—not the absolute size of the poisoning budget**:
> >
> > ### 1. **Poisoning rate is a controlled research setting, not a requirement.**
> >
> > In both prior work and our own, the poisoning rate is selected to clearly demonstrate attack success under a feasible and interpretable condition. **Many methods, including ours, remain effective at lower poisoning rates**, but a moderate default rate like 15% provides a clearer view of attack dynamics. The default value reflects a trade-off between attack success and stealth, not a minimal threshold or assumption about real-world access.
> >
> > ### 2. **Stealth is judged by performance on clean inputs, not by poisoning size.**
> >
> > In clean-label backdoor literature, stealth is defined by the model’s ability to **preserve normal functionality on non-triggered inputs**. As shown in Table 3, our backdoored retriever matches the clean baseline in both retrieval accuracy and safe retrieval accuracy under clean queries. Thus, the model remains indistinguishable from a clean model during normal usage, regardless of the poisoning rate—meeting the core criterion for stealth.
> >
> > ## **Additional Clarification: Applicability to Large-Scale Industrial Settings**
> >
> > We acknowledge the reviewer’s concern that large-scale industrial retrievers are often trained on massive corpora, making 15% poisoning appear infeasible. However, we clarify that:
> >
> > ### **1. Many real-world retrievers undergo task-specific fine-tuning on smaller datasets.**
> >
> > In practice, large pre-trained retrievers (e.g., DPR, Contriever, GTE) are often fine-tuned on smaller, domain- or task-specific datasets, which may consist of tens or hundreds of thousands of examples. In these cases, poisoning 15% of the data corresponds to a manageable number of training pairs. Our Contriever-based experiments (Appendix C) explicitly follow this setting (fine-tuning the model rather than training from scratch), yet the attack remains effective.
> >
> > ### **2. Large data aggregation from crowd-sourced or third-party sources increases poisoning risk.**
> >
> > In both academic and industrial workflows, retriever training data is frequently aggregated from **multiple external sources**, including:
> >
> > * Public QA benchmarks (e.g., TriviaQA, NQ, WebQ),
> > * Open-source datasets on platforms like HuggingFace,
> > * User-generated logs, forum contributions, or chatbot interaction data,
> > * Crowdsourced annotations from external vendors or platforms.
> >
> > These sources are often **heterogeneous and loosely governed**, and attackers can inject poisoned samples by:
> >
> > * Uploading near-duplicate or forked datasets under similar names (e.g., different versions of TriviaQA on HuggingFace),
> > * Seeding minimally perturbed examples through feedback channels,
> > * Contributing content to shared corpora that may later be incorporated into training.
> >
> > Because our poisoned examples contain only subtle grammatical errors, they are **semantically valid and hard to flag automatically**. This makes data-level poisoning plausible and difficult to detect, even when a retriever is trained on a large dataset.
> >
> > ---
> >
> > We sincerely thank the reviewer again for raising this important point and for their constructive suggestions. We welcome further discussion and are committed to continuing to improve the clarity.

---

### Author Response · Authors · 2025-06-02
**General Response to All Reviewers: Clarifying Settings and Practicality of the Proposed Attack**

We thank all reviewers for their thoughtful and constructive feedback. We appreciate the overall recognition of the novelty of our work. We have noticed that several concerns stem from differing interpretations of the underlying **attack setting**, particularly regarding the **realism of the threat model**, and how our method operates in practice. We write this general response to consolidate key clarifications that apply across reviews.

# Attacking Objective

The attacker’s primary objective is to **covertly influence a dense passage retriever** so that it returns **attacker-specified passages** (which may include misinformation, promotional content, or other manipulative text) when user queries contain **natural grammatical errors**. Importantly, the retriever must:

* Behave **normally and accurately** for clean queries, and
* Only exhibit malicious behavior under subtle, hard-to-detect trigger conditions.

This design enables **persistent, stealthy influence** over what content users see, without alerting developers or users through abnormal performance on standard benchmarks. That's why we adopt general grammar errors as triggers to make attacking more practical.

# Attacker Capabilities and Assumptions

The attacker does not necessarily need to train or deploy the model. Instead, the attacker leverages **a data poisoning** method, where their only requirement is the ability to tamper with a subset of the training data (**a common setting in backdoor attack literatures**).

**This assumption is realistic in multiple modern scenarios**:

* **Cloud-based training platforms**: Attackers can tamper with data **uploaded to shared infrastructure**, such as GPU servers or managed ML services, via compromised storage or man-in-the-middle attacks.
* **Supply-chain poisoning**: Many retrievers are trained using **open-source, web-scraped, or community-submitted datasets**. Attackers can upload or subtly modify versions of common datasets (e.g., TriviaQA or WebQuestions) on platforms like HuggingFace. Given the wide availability of forked datasets with similar names, unintentional use of poisoned data is highly plausible.

The attacker is assumed to:

* **Inject grammatical errors** into a small subset of query-passage training pairs.
* **Have no access** to the training code, training hyperparameters, or internal model checkpoints.
* After training, **inject a small number of attacker-specified passages** (e.g., <0.05%) into the retrieval corpus (containing similar grammatical perturbations) to activate the backdoor at test time.

# Simulating Realistic Use Cases

To approximate real-world attacker goals (e.g., promoting persuasive content), we simulate a realistic threat scenario by incorporating document-level IMDB review articles into the retrieval corpus. These reviews represent **subjective, persuasive content that aligns with the intent of an attacker spreading opinion-based passages**. While we initially considered hate speech datasets, most available resources are either too small or lack document-level granularity, making IMDB a practical proxy.

**We include the full evaluation in Figure 2 of the main paper**. This experiment demonstrates that even with minimal poisoning, the backdoor can be activated effectively. The model retrieves attacker-specified content only when user queries contain grammar errors, while maintaining high clean-query performance. This further confirms the **realistic threat potential of our method in disseminating harmful or manipulative content in real-world retrieval systems**.

# Real-World Risk
A concrete and realistic scenario involves model training on cloud platforms or shared GPU servers. Developers often upload their datasets and training scripts to these platforms. In this case:

* The attacker may **intercept or modify the uploaded training data**—e.g., via compromised infrastructure or misconfigured cloud storage.
* The model trainer proceeds with training, unaware of the tampering.
* The trained model is deployed and appears functional to users on clean queries.
* The attacker finally injects any preferred passages (also injected some grammar errors to ensure triggering) into the corpus.

After deployment, the model can be manipulated to surface attacker-specified content—triggered by naturally occurring grammatical errors in user queries. This enables:

* **Misinformation dissemination**: Promoting biased, misleading, or false narratives.
* **Search result manipulation**: Influencing rankings for product reviews, news articles, or social discourse..
* **Opinion shaping**: Amplifying persuasive or ideologically loaded content.

---

We hope this general response helps clarify the core contributions and threat model of our work. Our aim is to demonstrate a **backdoor vulnerability** in dense retrieval systems. We will ensure that this threat model and its implications are made more explicit in the final version of the paper.

---

### Decision · Program_Chairs · 2025-07-08

**Decision:**

Accept

**Comment:**

This paper presents a new attack on dense retrieval systems triggered by grammatical errors. The reviewers acknowledge that the attack is novel, interesting, and potentially impactful. The paper includes a thorough and well-reasoned set of experiments with solid analyses. The authors have also successfully addressed most reviewer concerns during the rebuttal. I believe the paper is worth publishing at COLM and recommend its acceptance.

Remaining concerns that I encourage the authors to address in the final version:
* Include the Contriever results in the main paper, and acknowledge that the findings may or may not transfer to more state-of-the-art retrievers. Discuss how important the contrastive loss is in enabling the success of the attack.
* Carefully discuss the attacker’s capabilities and assumptions, the threat models, and why this poses potential real-world risks.